# Therapeutic Textiles Functionalized with Keratin-Based Particles Encapsulating Terbinafine for the Treatment of Onychomycosis

**DOI:** 10.3390/ijms232213999

**Published:** 2022-11-13

**Authors:** André F. Costa, Salomé Luís, Jennifer Noro, Sónia Silva, Carla Silva, Artur Ribeiro

**Affiliations:** 1Centre of Biological Engineering, University of Minho, Campus de Gualtar, 4710-057 Braga, Portugal; 2LABBELS—Associate Laboratory, 4710-057 Braga, Portugal; 3INIAV—National Institute for Agrarian and Veterinarian Research, Rua dos Lagidos, 4485-655 Vairão, Portugal

**Keywords:** keratin-based particles, terbinafine, therapeutic textiles

## Abstract

Onychomycosis is the most common nail fungal infection worldwide. There are several therapy options available for onychomycosis, such as oral antifungals, topicals, and physical treatments. Terbinafine is in the frontline for the treatment of onychomycosis; however, several adverse effects are associated to its oral administration. In this work, innovative keratin-based carriers encapsulating terbinafine were designed to overcome the drawbacks related to the use this drug. Therapeutic textiles functionalized with keratin-based particles (100% keratin; 80% keratin/20% keratin-PEG) encapsulating terbinafine were developed. The controlled release of terbinafine from the functionalized textiles was evaluated against different mimetic biologic solutions (PBS buffer—pH = 7.4, micellar solution and acidic sweat solution—pH = 4.3). The modification of keratin with polyethylene glycol (PEG) moieties favored the release of terbinafine at the end of 48 h for all the solution conditions. When the activity of functionalized textiles was tested against *Trichophyton rubrum*, a differentiated inhibition was observed. Textiles functionalized with 80% keratin/20% keratin-PEG encapsulating terbinafine showed a 2-fold inhibition halo compared with the textiles containing 100% keratin-encapsulating terbinafine. No activity was observed for the textiles functionalized with keratin-based particles without terbinafine. The systems herein developed revealed therapeutic potential towards nail fungal infections, taking advantage of keratin-based particles affinity to keratin structures and of the keratinase activity of *T. rubrum.*

## 1. Introduction

Onychomycosis is the most common nail fungal infection worldwide, representing about 90% of toenail infections globally [1,2,3]. This infection, caused by dermatophytes, non-dermatophytes and yeast is characterized by local pain and paresthesia, hindering daily task performance and interfering with social interactions [1]. These fungi expresses proteases and keratinases to breakdown keratin, leading to the colonization, invasion, and infection of the skin’s stratum corneum, hair shafts, and nails [4]. Onychomycosis can occur at any age, although it is more prevalent in the elderly [1,5]. Those with a history of nail damage, psoriasis, diabetes, poor peripheral circulation, HIV, immunosuppression, and smokers are considered more vulnerable [1,6]. The exposure of the nail to warm and moist environments, such as occlusive footwear, as well as genetic susceptibility can facilitate the infection and the spread of the disease.

*Trichophyton rubrum* is a anthropophilic filamentous fungus that infects keratinized structures such as skin, hair, and nails, being the primary cause of tinea pedis, tinea corporis, and onychomycosis [7,8]. It has a global distribution and has become an increasingly common dermatophyte since the 1950s, following a change in habits, such as the use of occlusive footwear. This fungus is characterized morphologically by rapid growth and high sporulation in culture [7,9].

There are various therapy options available for onychomycosis, such oral and topical antifungals, and physical treatments. In the United States, until 2019, only three onychomycosis topical treatments were approved by the FDA, ciclopirox 8% nail lacquer (also approved for commercialization in Europe) [10], efinaconazole 10% solution, and tavaborole 5% solution [1]. Efinaconazole is a triazole that inhibits lanosterol 14a-demethylase and is effective against dermatophytes, such as *Candida* spp. [1]. Tavaborole is a benzoxaborole that inhibits protein synthesis via fungal aminoacyl transfer RNA synthetase with a broad antifungal activity [1]. Burning, itching, and stinging at the application site are some of the possible side effects of these treatments [10]. Laser and photodynamic therapy are two physical therapies that have grown in popularity as a result of the success of in vitro studies. Moreover, FDA has approved several neodymium: yttrium aluminum-garnet (Nd: YAG) laser therapies for the treatment of onychomycosis [10].

The most commonly used oral medications for the treatment of onychomycosis are antifungals of the azole (itraconazole, fluconazole, and ketoconazole) and allylamine (terbinafine) families. Headaches, taste disturbances, dermatitis, anorexia, vomiting, epigastric pain, diarrhea, drug-to-drug interactions, and, in rare cases, depression, neutropenia, and hepatic dysfunction are all possible side effects of these oral treatments [10,11]. Although oral antifungal therapies are effective, their use is limited by the significant adverse effects and the possibility of systemic toxicity and drug–drug interactions, making topical antifungals an appealing solution for the treatment of onychomycosis [11,12].

On a transungual drug delivery system the main route for drug penetration is through the nail plate, however its architecture and constitution restricts the effective delivery of therapeutic anti-fungal concentrations at the infection site [13]. Other factors, like pH, molecular size, and nature of the vehicle can also affect the transport of drugs through the nail [13]. The ability of a molecule to cross the nail keratin network is reduced as the molecular size of the therapeutic agent increases [13,14]. The pH of the formulation appears to also have a significant influence on the penetration through the nail plate. Uncharged species (neutral pH) penetrate to a higher extent than charged ones (acid and basic pH) [13]. The water content of a therapeutic formulation greatly influences the degree of molecules penetration through the nail. Given that the nail plate acts as an keratin-based hydrogel, its swelling enhances the permeation of the therapeutic molecules to the infection site [13].

Topical antifungal penetration therapies require the use of a vehicle particularly designed for transungual delivery [13]. To effectively penetrate the nail, the design of such a vehicle must take into account nail properties (thickness, hydration) and drug physicochemical properties (size, shape, hydrophobicity) [14].

Terbinafine is an allylamine efficient against dermatophytes that inhibits the enzyme squalene epoxidase, preventing the development of functional fungal cell membranes [1,15,16]. For fingernail and toenail infections, a typical oral treatment regimen is 250 mg twice a day for 6 to 12 weeks. According to several studies, terbinafine has a better likelihood of curing onychomycosis than azoles and griseofulvin, with the same or less side effects [2,17]. The mycological cure rate for terbinafine in toenails is 70%, while the total cure rate is 38 % [1,3].

Aiming to circumvent the drawbacks associated to the hepatotoxicity of the oral therapies and the low drug delivery rates of topical approaches, we designed a new keratin-based carrier system for the local delivery of terbinafine.

Keratin, is the most abundant component of wool, hair, nails, and hooves [18,19]. This protein shows great biocompatibility in a wide range of biomedical and biotechnological applications, including films [20,21,22], sponges [23], and fibers [24,25]. Recently, keratin-based drug carriers, such as nanoparticles [26,27], micelles [28], nanogels [29,30], and films [31], have also been explored. Keratin can be easily extracted from feathers [32] and human hair [19]. These are common waste products from the poultry industry and hair salons, respectively. Moreover, when developing formulations with keratin extracted from human hair we can explore the natural affinity of this protein towards keratinaceous appendices [19].

In this work, terbinafine carriers based on keratin and PEGylated keratin were designed for the treatment of onychomycosis. A therapeutic textile, functionalized with the keratin-based particles encapsulating terbinafine, was obtained and further characterized. This device is presented herein as a new wearable and comfortable strategy for the local treatment of onychomycosis.

## 2. Results and Discussion

### 2.1. Keratin PEGylation

Proteins have recently gained popularity due to their promising properties, which include high biocompatibility, biodegradability, and drug binding capacity [33,34]. The use of hair keratin as a core material to produce particles herein, is derived from its great biocompatibility and intrinsic affinity towards keratin structures, including nail plate keratins [19]. Several studies demonstrated that nanoparticles PEGylation generally confer protection (“corona effect”), thus avoiding aggregation, opsonization, and phagocytosis [35]. PEG coatings can improve biological barrier penetration by reducing interactions with extracellular matrix, cellular barriers, and biological fluids, resulting in improved delivery while increasing particle stability through steric repulsion [35,36].

The keratin was PEGylated using a monofunctional PEG containing an aldehyde group (MW 5 kDa), as described in the literature [37]. The reaction took place at an acidic pH (5.1) and in the presence of a reducing agent (NaBH3CN) (Figure 1).

The PEGylation degree of keratin was measured by the quantification of the free amine residues (non-PEGylated) by the TNBSA assay. This technique is based on the interaction of 2,4,6-trinitrobenzene sulfonic acid (TNBSA) with the free amine residues of keratin. The total number of lysine residues accessible in each protein gave us indication of a PEGylation degree around 70%, containing 0.404 mg _keratin_/mg _PEGylated keratin_.

### 2.2. Characterization of the Keratin-Based Particles

The particles formation efficiency was greater than 90% for all the keratin-based particles, regardless of the presence of terbinafine. Moreover, the particles’ formation efficiency observed for the samples with terbinafine (90.1 ± 0.1 for 100% keratin-Terb particles and 93.5 ± 0.4 for 80% keratin/20% keratin-PEG-Terb) indicate that the presence of terbinafine did not affected the formation process of keratin particles.

The high encapsulation efficiencies of terbinafine were also observed for the 100% keratin-Terb particles (99.80 ± 0.10%) and for 80% keratin/20% keratin-PEG-Terb particles (99.90 ± 0.50 %) (Table 1). These values were greater than the reported for terbinafine encapsulation in other studies (39.50%) with the same production methodology [38].

The physio-chemical characterization revealed particles with a size diameter between 500 and 756 nm, a PDI (polydispersity index) between 0.123 and 0.292 and a negative surface charge (≈−30 mV) (Figure 2). One can observe that although the particles encapsulating terbinafine present higher Z-Average when compared with the keratin particles without the drug encapsulated, they also show a higher stability over time suggesting that terbinafine can increase the keratin-based particles stability. Although the size of the nanoparticles is statistically different between week 0 and week 14, no variation was observed for the PDI and the surface charge, confirming the stability of the nanoparticles [39].

The size of the keratin particles herein developed (≈600 nm) is higher than the most nanoparticles described in the literature for topical usage (≈100–500 nm) [40,41,42,43]; however, it is still the ideal for this particular application, allowing fabric coating rather than penetration into the fabric fibers [44]. This larger size results in a higher surface coating area, which might favor the release of terbinafine [45].

The shape and the size of the keratin-based particles were also evaluated by scanning electron transmission microscopy (STEM). While analyzing STEM micrographs (Figure 3) of selected samples, it was possible to confirm the particles’ spherical shape and size previously analyzed by dynamic light scattering.

FTIR analysis was used to analyze the intermolecular interactions between keratin and PEGylated keratin during particle formation, as well as between the terbinafine and the keratin of the particles (Figure 4).

The spectra of keratin and PEGylated keratin revealed the typical protein peaks, corresponding to the amide I and amide II (C = O bonds). These peaks, amide I and amide II, occurred at 1638 and 1523 cm^−1^ for both keratins. Peaks at 634, 1024, and 3265 cm^−1^, corresponding to C–S bonds in cystine, C–O stretching vibrations, and N–H bonds, were also observed for both keratins. Comparing the spectra of the keratins with the spectra of the particles, the appearance of two new peaks at 2850 and 2920 cm^−1^ was observed (* in Figure 4). These peaks may be related with the interaction between the proteins during the particle’s formation. The FT-IR spectrum of pure terbinafine hydrochloride shows C=C stretching bands at 1517 cm^−1^, aromatic C≡C stretching bands at 2218 cm^−1^, aromatic C-H stretching bands at 3044 cm^−1^, aromatic alkenyl C=C-H stretching bands at 2968 cm^−1^, C-N bands at 1133 cm^−1^, and aliphatic C-H stretching bands at 2570 cm^−1^ [46]. Most of these peaks were not detected because they can be masked when terbinafine is encapsulated onto the keratin particles.

#### Characterization of the Functionalized Cotton Fabrics

After cotton fabrics, the functionalization of weight gain was inferred to determine the amount of material coated at fabrics surface (Table 2). Regardless of the keratin particle composition, the weight gain was similar for all the conditions tested, confirming the ability of the developed systems to be applied on cotton fabric functionalization.

Scanning electronic microscopy (SEM) was used to examine the surface morphology of cotton samples functionalized with the keratin-based particles. The non-functionalized cotton textiles (control) had a smooth and uniform surface, while the functionalized fabrics presented particle aggregates at the surface of the fibers (shown by the black arrows in Figure 5). The presence of the terbinafine, encapsulated onto keratin particles, did not disturb the particles’ deposition pattern (Figure 5).

The thermogravimetric studies of functionalized samples were carried out to determine the effect of coating with the keratin-based particles on the thermal stability of the cotton fabrics (Figure 6). The thermogravimetric analysis show that all the textiles suffer an initial water loss between 99.90 and 158.80 °C. After that it is observable that the degradation of the cotton fabrics functionalized with keratin-based particles start at lower temperatures (between 213.40 °C and 348.30 °C) than the raw fabric (440.60 °C). The olive oil present in the particle composition, with associated degradation temperatures between 200 °C and 420 °C, is responsible for the samples weight loss [47]. The major weight loss (53.40 °C–78.40%) of the functionalized fabrics is observed for temperatures ranging from 458.70 °C to 545.70 °C. Comparing the degradation profiles of the functionalized and raw fabrics it is clear an improvement of the resistance to thermal degradation with the best results observed for 80% keratin/20% keratin-PEG-Terb.

The effect of media (PBS (pH = 7.4), acid sweat (pH = 4.3), and micellar solution) on the release of terbinafine from the functionalized cotton fabrics was investigated (Figure 7). The conditioning media were chosen to mimic the release during perspiration (sweat) and to mimic the pH (PBS) [44] and the membranous/aqueous environment encountered in the interlamellar regions of the *Stratum corneum* (micellar solution) [48].

After 48 h, the release of terbinafine from the functionalized cotton fabrics was more efficient in PBS buffer than in the micellar or acidic sweat solutions (Figure 7). For the micellar solution, no release was observed from the cotton fabrics functionalized with 100% keratin-Terb particles, while for the cotton fabrics functionalized with 80% keratin/20% keratin-PEG-Terb, a maximum release of 3.00 µg/mL was observed. When the release was performed against the acidic sweat solution (pH = 4.3), the presence of the PEGylated keratin promoted the release of terbinafine. While the fabrics functionalized with 100% keratin-Terb particles presented a release of 0.93 µg/mL at the end of 8 h, the release from the fabrics containing 80% keratin/20% keratin-PEG-Terb particles were 20.85 µg/mL. The same trend was also observed for the samples incubated in PBS buffer. A terbinafine release of 538 µg/mL and of 683.00 µg/mL was observed at the end of 8 h for the 100% keratin-Terb and for the 80% keratin/20% keratin-PEG-Ter functionalized cotton fabrics, respectively. These results were expected, since the PEGylation of the keratin particles can favor drug release [36]. The presence of PEG at the core of the particles may form water channels, favoring drug diffusion and release [36].

The terbinafine release from the cotton fabrics functionalized with the keratin-based particles was comparable with the outcomes in the literature for other systems encapsulating this drug [49,50]. There was an increase in the amount of terbinafine released, with the time of incubation, until it reached a plateau phase where the terbinafine concentration in the medium was maintained, remaining almost unchanged [49,50]. For the cotton fabrics with the keratin-based particles, the plateau phase was reached at around 8 h of incubation regardless of the particle composition and incubation media (PBS, micellar solution and sweat acidic solution). However, it is noteworthy that the release of terbinafine from the cotton fabrics functionalized with the keratin-based systems was not as fast as for the other encapsulating systems described in the literature [51]. The differences observed were expected, since we were comparing the release of terbinafine from functionalized textiles (entrapped particles) with the release from free particles. This slower release observed for the functionalized cotton fabrics with the keratin-based particles encapsulating terbinafine is ideal for the treatment of onychomycoses. Longer contact times and a sustained controlled released of the drug are key parameters for an efficient treatment of the affected areas of the nails [14].

Air permeability is commonly described as the quantity of airflow that passes through a given area of a fabric. Because this characteristic has a significant impact on the textile’s thermal comfort, the capacity to be permeable to air may alter the wearability of a specific textile product [52]. The effect of keratin particles coating on the air permeability of the functionalized cotton fabrics was evaluated (Figure 8).

The air permeability of the raw cotton fabric (268 1/m^2^s) is similar to the described in others works (223; 234 1/m^2^s) [44,53]. Furthermore, the functionalized fabrics presented higher air permeability than the non-functionalized sample (raw). Moreover, the functionalized fabrics with the particles encapsulating terbinafine have a significant increase in air permeability than their counterparts. Despite air permeability normally decreasing with the coating of textiles surface with nanoparticles [54,55], these results show that the cotton fabrics functionalized with the keratin-based particles are highly permeable to air. This high permeability represent an advantage compared with other systems, as it could result in higher comfort [52] and the wearability of the functionalized cotton fabrics in the context of the treatment of the onychomycosis.

### 2.3. Antimicrobial and Antifungal Activity

The ability of the keratin-based particles encapsulating terbinafine to confer antimicrobial and antifungal activity to the cotton textiles was evaluated by using a qualitative method (Figure 9). The activity of the samples functionalized with keratin particles with and without terbinafine was tested against *Trichophyton rubrum* (fungus), *Candida parapsilosis* (yeast), *Escherichia coli* (Gram-negative bacteria), and *Staphylococcus aureus* (Gram-positive bacteria). According to the findings (Figure 9), the textiles functionalized with the keratin particles containing terbinafine presented excellent antifungal activity against *Trichophyton rubrum*. Textiles functionalized with 80% keratin/20% keratin-PEG encapsulating terbinafine showed an approximately 2-fold inhibition halo (25.00 ± 3.00 mm) compared with the textiles containing 100% keratin-encapsulating terbinafine (14.70 ± 0,30 mm) (Figure 9). The antifungal activity was enhanced by the presence of the PEGylated keratin on the particles’ composition, as higher growth inhibition can be observed for the cotton samples functionalized with the 80% keratin/20% keratin-PEG-Terb particles. These results agree with the observed for the release studies, where the presence of the PEGylated keratin increased the amount of terbinafine released along time. The antifungal activity against *T. rubrum* was only associated with the presence and release of terbinafine [16], as no inhibition was observed for the samples functionalized with the keratin particles without the drug, indicating that neither the keratin or the olive oil present activity against this fungus. Since this fungus is responsible for the appearance and development of onychomycosis in the nails, this fabrics can be used in the treatment of the disease [11]. In this study no effect was observed for the *C. parapsilosis, E. coli,* and *S. aureus* (Appendix A).

As previous mentioned, *T. rubrum* produced protease and keratinase to breakdown keratin, leading to a colonization, invasion, and infection of the nails [4]. Since these particles are made of keratin, the keratinases produced by the fungi degraded the keratin of the particles, promoting the release of the encapsulated drug (terbinafine). Herein we take advantage of the enzymatic machinery of the fungus to trigger the release of the drug from the developed system.

## 3. Materials and Methods

Terbinafine hydrochloride was purchased from TCI chemicals, Belgium. A local hairdresser provided the hair samples used for keratin extraction. The fabric use in this work was provided the company MGC-AT. The olive oil used in the nanoparticles formation is a cooking olive oil purchased from a local supermarket. Ethanol HPLC grade was obtained from Fluka, Portugal. All other reagents were acquired from Merck Sigma, Spain, and used as received.

### 3.1. Keratin Extraction and Purification

Keratin was isolated from human hair. The removal of pollutants and lipids from hair was performed according to the IAEA/RL/50 1978 requirements, and the keratin was extracted using a procedure adapted from Ayutthaya et al., 2015 [56]. The hair (1:10; grams of dried hair to mL of solution) was incubated in a solution containing 8 M urea, 0.2 M SDS, and 0.5 M sodium metabisulphite. The mixture was heated to 100 °C for 30 min before incubation overnight at 37 °C under agitation. After centrifugating the extraction solution at 2800 g for 10 min, the supernatant was filtered to eliminate hair debris. The keratin solution was then dialyzed against distilled water for 5 days at room temperature using a dialysis membrane with a 14 kDa cutoff. [39,57]

### 3.2. Keratin PEGylation

The keratin was PEGylated according to the method described by Mayolo-Deloisa et al. [37] (Figure 1). The Keratin solution (37.3 mg/mL) was prepared with phosphate buffer (100 mM phosphate + 20 mM NaBH3CN) at pH = 5.1, in a 2:5 ratio (protein: buffer), and 200 mg of the mono functional PEG (aldehyde function, MW 5 kDa) were added to this solution, under stirring. After incubation overnight at 4 °C at 700 rpm the solution was ultra-filtrated using a membrane of 10 kDa for the removal of unreacted PEG and buffer. The PEGylated keratin was recovered as a white solid after freeze-drying. The degree of keratin PEGylation was evaluated using a colorimetric titration approach through the reaction of 2,4,6-trinitrobenzene sulfonic acid (TNBSA) with the free amine residues of keratin. The protein concentration was quantified using the DC protein assay (BIO-RAD), following the manufacturer instructions [34].

### 3.3. Production of Keratin-Based Particles

#### Particles Formation

The solutions of 10 g/L of keratin (100% keratin) and of 8 g/L keratin with 2 g/L PEGylated keratin (80% keratin/20% keratin-PEG) were prepared in water. A stock solution of terbinafine (40mg/mL) was prepared in 100% ethanol, and 100 µL of this solution was added to 400 µL of olive oil until complete dissolution. The particles containing the antifungal agent (100% keratin-Terb; 80% keratin/20% keratin-PEG-Terb) were prepared by adding a 0.5 mL of terbinafine organic solution to 9.5mL of aqueous keratin solution (final terbinafine concentration of 0.4 mg/mL). The final solution was subjected to ultrasonication for 6 min (8s ON/2s OFF cycles). The keratin particles without terbinafine (100% keratin; 80% keratin/20% keratin-PEG) were prepared by adding 0.5 mL of olive oil to 9.5 mL of keratin solutions using the same methodology.

### 3.4. Particles Characterization

#### 3.4.1. Formation Efficiency

The efficiency of particles formation was determined by separating the free protein from formed particles using 100 kDa centrifuge Amicon tubes. The DC^TM^ colorimetric technique (detergent compatible) was used to quantify the free protein. The absorbance of the samples was measured at 750 nm and the free keratin concentration was determined against a calibration curve of BSA standard solutions (0.05 mg/mL–1.5 mg/mL). The formation efficiency was calculated using the following Equation (1), were [Protein]_initial_ is the initial amount of keratin and [Protein]_free_ is the free keratin.
(1)$$\mathrm{Formation}\;\mathrm{Efficiency}(\%)=\frac{{\left[\mathrm{Protein}\right]}_{\mathrm{initial}}\;-\;{\left[\mathrm{Protein}\right]}_{\mathrm{free}}}{{\left[\mathrm{Protein}\right]}_{\mathrm{initial}}}\;\times \;100$$

#### 3.4.2. Encapsulation Efficiency

The efficiency of encapsulation was evaluated by the separation of the free terbinafine using a PD-10, Sephadex TM G-25 column. The quantification of the free terbinafine was made by UV–Visible spectroscopy (λ = 285 nm) against a calibration curve of different concentrations of terbinafine prepared in water (between 0 and 100 µg/mL). The encapsulation efficiency was calculated using the Equation (2), where [Terbinafine]_initial_ is the initial amount of drug and [Terbinafine]_free_ is the free amount of terbinafine.
(2)$$\mathrm{Encapsulation}\;\mathrm{Efficiency}=\frac{{\left[\mathrm{Terbinafine}\right]}_{\mathrm{initial}}\;-\;{\left[\mathrm{Terbinafine}\right]}_{\mathrm{free}}}{{\left[\mathrm{Terbinafine}\right]}_{\mathrm{initial}}}\;\times \;100$$

#### 3.4.3. Particles size and Surface Charge

The mean size diameter, polydispersity index (PDI), and surface charge (ζ) of keratin-based particles were determined using a Zetasizer Nano ZS (Malvern Instruments). The samples were measured using a 1:10 dilution and read in triplicate, with the data expressed as mean ± standard deviation.

#### 3.4.4. Fourier Transform Infrared Spectroscopy

Fourier Transform Infrared Spectroscopy (FTIR) was used to investigate changes in the chemical structure of keratin-based particles. The spectra were collected using a Bruker Alpha II (Billerica, MA, USA) using the Opus 8.22.28 software. Samples were placed directly on the crystal, and spectra were collected between 400 and 4000 cm^−1^ wavenumbers at a resolution of 2 cm^−1^.

#### 3.4.5. Scanning Electron Transmission Microscopy (STEM)

The morphology of keratin-based particles was evaluated by STEM analysis. The diluted particles suspensions were dropped on copper grids with a 400-mesh carbon film, 3mm in diameter.

The shape and morphology of the particles were observed using a NOVA Nano SEM 200 FEI instrument.

#### 3.4.6. Functionalization of Fabrics with Keratin-Based Particles

Prior to functionalization, cotton fabrics (12 columns/cm; 25.18 g/m^2^) were washed with a 0.1% Lutensol AT 25 solution for 1 h at 50 °C, rinsed with deionized water, and dried in an oven at 40 °C. Cotton fabrics were padded with keratin-based particles using 30 cycle passages (P = 1.5 Bar; v= 2.5 m/min). The padding conditions were previously optimized by padding cotton fabrics with water until 70% weight gain.

The percentage of weight gain after functionalization with keratin-based particles was obtained by calculation using Equation (3), where Weight _Final_ is the weight of the fabric after functionalization and Weight _Initial_ is the weight of the fabric before functionalization.
(3)$$\%\;\mathrm{Weight}\;\mathrm{Gain}=\frac{{\mathrm{Weight}}_{\mathrm{Final}}{\;-\;\mathrm{Weight}}_{\mathrm{Initial}}}{{\mathrm{Weight}}_{\mathrm{Initial}}}\;\times \;100$$

### 3.5. Characterization of Functionalized Fabrics

#### 3.5.1. Scanning Electron Microscope

A desktop scanning electron microscope (SEM) was used to characterize the functionalized fabrics (Phenom-World BV, Eindhoven, The Netherlands). The ProSuite program was used to collect all the data. The functionalized fabrics were adhered onto aluminum pin stubs using electrically conductive carbon adhesive tape (PELCO TabsTM) and the observation was realized at 10 kV [58].

#### 3.5.2. Air Permeability

The air permeability of the functionalized fabrics was quantified using the standard technique NP EN ISO 9237. Samples with a 20 cm^2^ area were tested using a vapor pressure of 100 Pa. For each sample, all tests were performed ten times.

#### 3.5.3. Thermogravimetric Analysis

Thermogravimetric studies were carried out in a TGA 4000 (Perkin Elmer, Waltham, MA, USA) with a 9 mg sample weight alumina crucible. Curie temperatures of reference for Alumel, Nickel, and Perkalloy were used to calibrate the temperature at the same sweep rate as the sample. The samples were analyzed at temperatures ranging from 25 to 800 °C at a rate of 20 °C/min in a nitrogen environment (flow rate: 20 mL/min). The percentage of weight loss and its derivatives were plotted versus temperature in Piris software.

#### 3.5.4. Release of Terbinafine from Functionalized Textiles

To evaluate the terbinafine release through time a functionalized sample (1 cm × 1cm) was placed inside a 1 kDa dialysis tube and the release was evaluated in PBS buffer (pH = 7.4); micellar solution (16 mM sodium dodecyl sulfate) and acidic sweat solution (10 g/L NaCl, 1 g/L 85% Latic acid, 0.25 g/L L-histidine monohydrochloride, 1 g/L sodium phosphate dibasic anhydrous; pH = 4.3)). The absorbance at 285 nm was registered at 1, 2, 4, 8, 24, and 48 h of release. The amount of terbinafine release was quantified against a calibration curve (0–0.75 mg/mL).

### 3.6. Antimicrobial and Antifungal Activity

The antimicrobial activity of the functionalized fabrics with 100% keratin-Terb; 80% keratin/20% keratin-PEG-Terb, and the controls without encapsulated terbinafine (100% keratin and 80% keratin/20% keratin-PEG), were tested against two species of fungi (*Trichophyton rubrum* MUM 0805 and *Candida parapsilosis* ATCC 22019) and two species of bacteria (*Escherichia coli* ATCC 6538-Gram negative and *Staphylococcus aureus* CECT 434-Gram positive). To evaluate the antimicrobial activity, a method based on ISO 20645/2005 was adapted. The bacteria were inoculated in 30 mL of Tryptic Soy Broth (TSB) and the yeast in 30 mL of Sabouraud Dextrose Broth (SDB), both were incubated for 18 h at 37 °C, with a rotation frequency of 120 rpm. After the incubation period, the inoculum was adjusted to an optical density (OD) below 0.5 at 620 nm and properly diluted to 1 × 10^6^ CFU mL^−1^ for bacteria, while for the yeast, the cell density was adjusted to 1 × 10^5^ CFU mL^−1^. An aliquot of cellular suspension (100 μL) was spread in Tryptic Soy Agar (TSA) and Sabouraud Dextrose Agar (SDA) petri dishes to bacteria and yeast, respectively.

The *T. rubrum* was seeded on 2 mL of sloping potato dextrose agar (PDA, Frilabo) at room temperature for 7 days. Afterward, 3 mL of distilled water was added and shaken, and the cell density was adjusted to the optical density (OD) of 0.08–0.1 at 620 nm and properly diluted to 1 × 10^5^ CFU mL^−1^. The samples were then placed on two-layer agar plates. The lower layer consisted of 10 ± 0.1 mL culture medium potato dextrose agar (PDA, Frilabo, Maia, Portugal) free from fungi and the upper layer was inoculated with fungal culture (1 × 10^7^ CFU mL^−1^). The plates were shaken vigorously to distribute fungi evenly, and then incubated at room temperature for approximately 7 days. Afterwards the inhibition halos were measured and recorded using Image LabTM software.

All experiments were carried out in triplicate and repeated in at least three independent assays.

### 3.7. Statistical Analysis

Statistical comparisons were performed by one-way ANOVA with GraphPad Prism 5.0 software (La Jolla, San Diego, CA, USA). Tukey’s post hoc test was used to compare all the results between them. A *p*-value ≤ 0.05 was considered statistically significant.

## 4. Conclusions

We successfully developed functionalized cotton textiles with keratin-based particles encapsulating terbinafine for the treatment of onychomycosis. All the particles developed had a formation efficiency greater than 90% and the particles with terbinafine presented encapsulation efficiency greater than 99.8%. The controlled release of terbinafine from the textiles coated with keratin particles was tested against different mimetic biologic solutions. The release profile was dependent on the external media used during dialysis (PBS buffer and sweet and micellar solution) and the presence of PEGylated keratin in the nanoparticle constitution. The fabrics functionalized with 80% keratin/20% keratin-PEG-Terb particles showed the highest terbinafine release (20.85 µg/mL and 683.00 µg/mL in acid sweat and PBS, respectively). The combination of keratin-based particles, target for keratinases, and terbinafine, an antifungal agent, made possible the development of a new therapeutic textiles for the treatment of onychomycosis with high antifungal activity against *Trichophyton rubrum*. Other strategies, such as films and nail lacquers, supplemented with the keratin-based particles encapsulating terbinafine, can also be explored in the future.

## Figures and Tables

**Figure 1 ijms-23-13999-f001:**
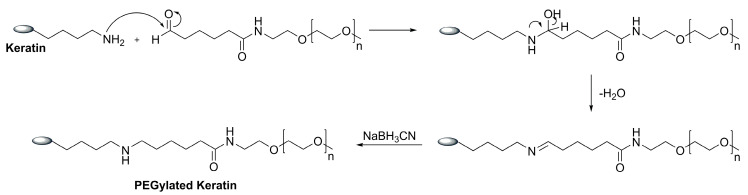
Reactional scheme of keratin PEGylation.

**Figure 2 ijms-23-13999-f002:**
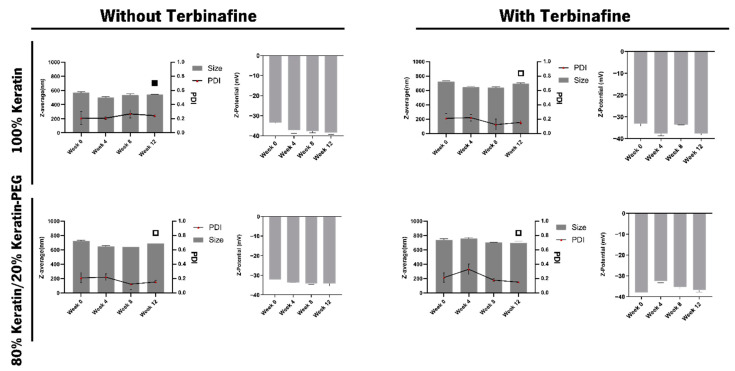
Stability characterization of keratin-based particles, with and without terbinafine, during storage at 4 °C. The data represent the mean ± SD of the Z-Average (nm), PDI and surface charge (z-Potential (mV)) for 14 weeks. The data were analyzed by Tukey’s multiple comparison test. Samples marked with ▪, ▫ show significative differences (*p*-value 0.001 (▫), *p*-value 0.0001 (▪)) when compared with week 0.

**Figure 3 ijms-23-13999-f003:**
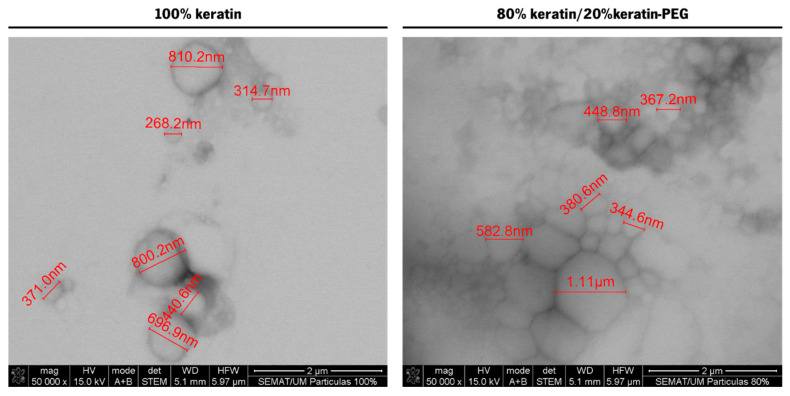
STEM micrographs (50,000×) of 100% keratin and 80% keratin/20% keratin-PEG particles.

**Figure 4 ijms-23-13999-f004:**
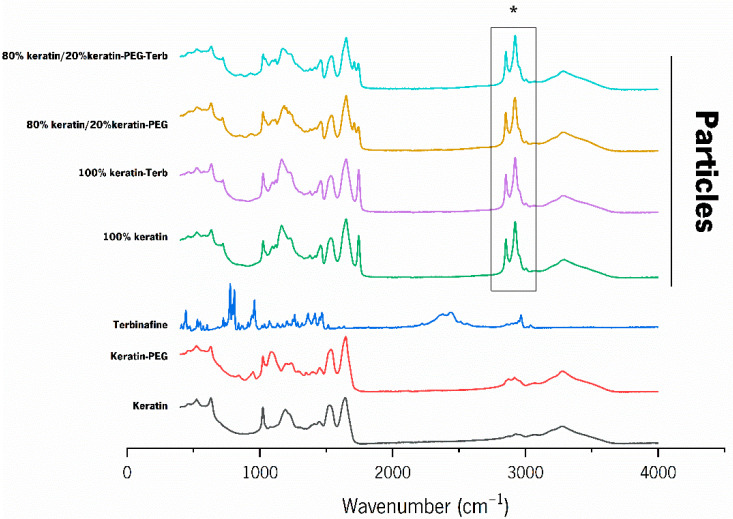
FTIR spectra of keratin, PEGylated keratin (Keratin-PEG), terbinafine and keratin particles (100% keratin; 100% keratin-Terb; 80% keratin/20% keratin-PEG; 80% keratin/20% keratin-PEG-Terb). * Corresponds to new peaks (2850 and 2920 cm^−1^) after particles formation.

**Figure 5 ijms-23-13999-f005:**
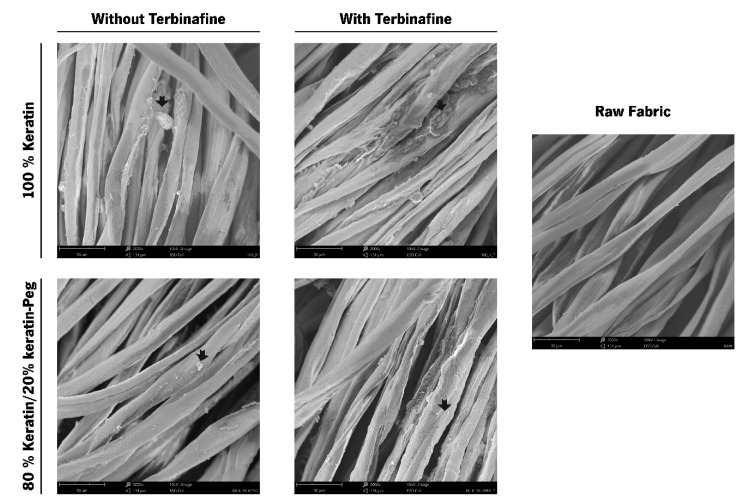
SEM images of cotton fabrics functionalized with keratin particles (100% keratin; 100% keratin-Terb; 80% keratin/20% keratin-PEG; 80% keratin/20% keratin-PEG-Terb) and control sample (raw cotton fabric); images were obtained with an amplification magnitude of 2000×. The arrows highlight the coating with keratin-based particles at the fibers and at the inter-fiber spaces.

**Figure 6 ijms-23-13999-f006:**
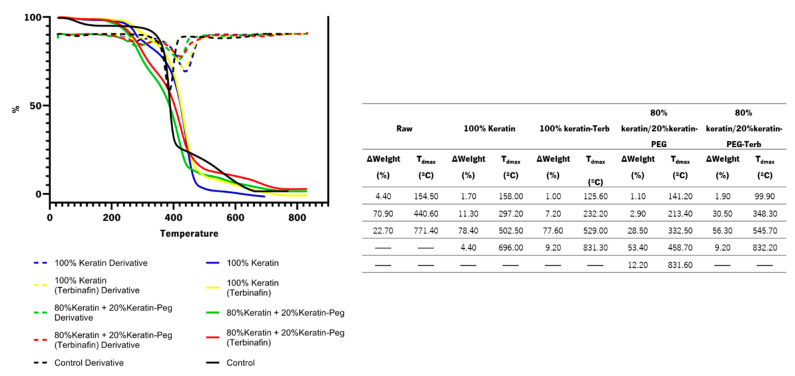
TGA profiles of cotton fabrics functionalized with keratin particles (100% keratin; 100% keratin-Terb; 80% keratin/20% keratin-PEG; 80% keratin/20% keratin-PEG-Terb). The control corresponds to the cotton fabric without keratin-based particles. The thermogravimetric analysis of cotton fabrics (control) and fabrics functionalized with keratin-based particles (∆Weight (%) represents weight loss and T_dmax_ the time-point temperature).

**Figure 7 ijms-23-13999-f007:**
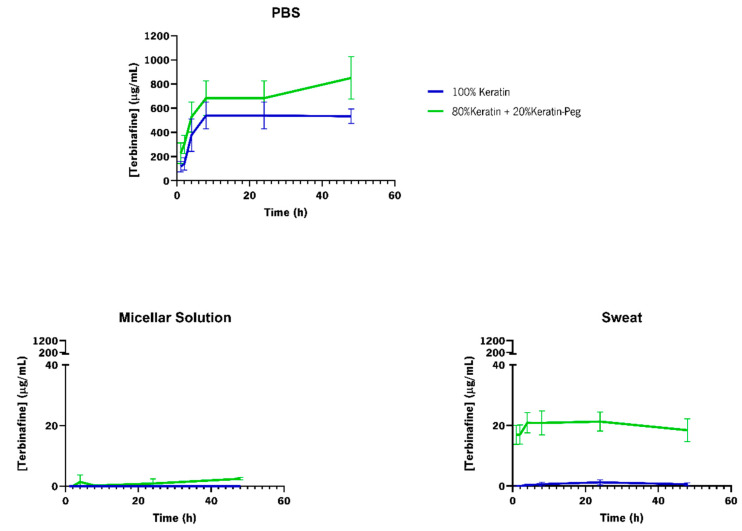
Terbinafine cumulative controlled release profiles from functionalized cotton fabrics with keratin-based particles encapsulating terbinafine. The release was performed under different conditions (against PBS (pH 7.4), micellar solution and acidic sweat solution (pH 4.3)), 100% keratin-Terb particles (blue), and 80% keratin/20% keratin-PEG-Terb (green).

**Figure 8 ijms-23-13999-f008:**
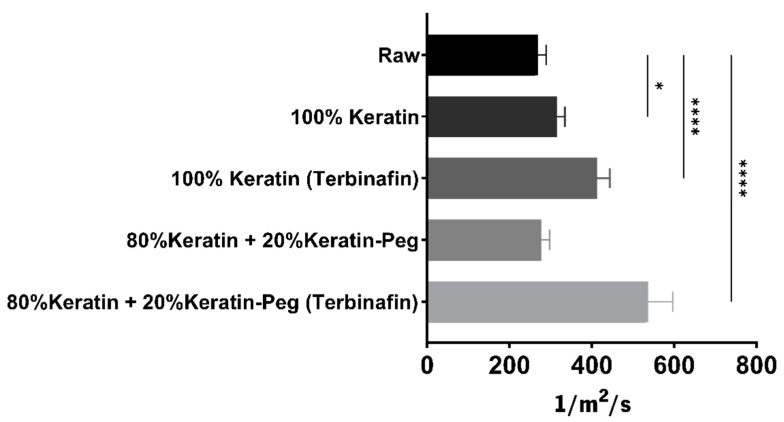
Air Permeability of functionalized fabrics with keratin particles (100% keratin; 100% keratin-Terb; 80% keratin/20% keratin-PEG; 80% keratin/20% keratin-PEG-Terb). The data were analyzed by Tukey’s multiple comparison test: the samples marked with * and **** represent significant differences (*p*-value ≤ 0.05 (*) and *p*-value ≤ 0.0001 when compared with the raw sample).

**Figure 9 ijms-23-13999-f009:**
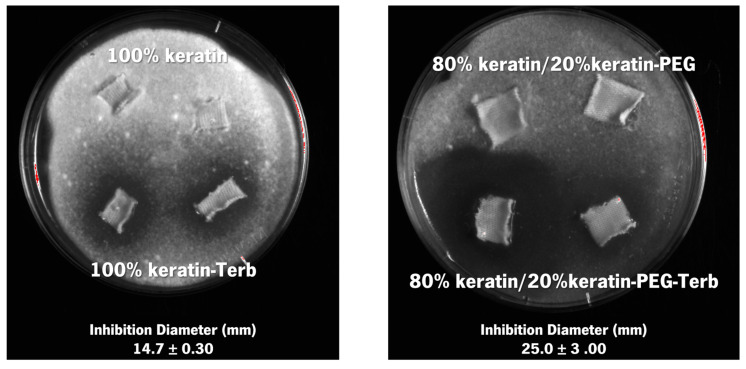
Antifungal activity of the functionalized fabrics (100% keratin; 100% keratin-Terb; 80% keratin/20% keratin-PEG; 80% keratin/20% keratin-PEG-Terb) against *Trichophyton rubrum*.

**Table 1 ijms-23-13999-t001:** Formation and encapsulation efficiency (%) of keratin-based particles with and without terbinafine.

Particles	Formation Efficiency (%)	Encapsulation Efficiency (%)
100% keratin	90.70 ± 0.60	--------
100% keratin-Terb	90.10 ± 0.10	99.8 ± 0.10
80% keratin/20% keratin-PEG	92.30 ± 0.20	--------
80% keratin/20% keratin-PEG-Terb	93.50 ± 0.40	99.9 ± 0.50

**Table 2 ijms-23-13999-t002:** Weight gain (%) of the cotton fabrics after functionalization with keratin-based particles (with and without terbinafine).

Fabric	Weight Gain (%) (±SD)
100% keratin	5.07 ± 0.03
100% keratin-Terb	5.03 ± 0.07
80% keratin/20% keratin-PEG	4.31 ± 0.23
80% keratin/20% keratin-PEG-Terb	5.59 ± 0.05

## Data Availability

Not applicable.

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
