# Peer review of "Therapeutic Textiles Functionalized with Keratin-Based Particles Encapsulating Terbinafine for the Treatment of Onychomycosis"

_ijms, 2022, doi:10.3390/ijms232213999_

Round 1

Reviewer 1 Report

The article Therapeutic textiles functionalized with keratin-based particles encapsulating terbinafine for the treatment of Onychomycosis is a very interesting proposal that would help a very common problem. He only sent some comments and doubts.

• Review the writing because a text repeatedly appears indicating that there is an error.

• Because the authors carried out keratin extraction from hair and did they not use a reactive grade compound (to ensure composition)? It may be that the process that the authors carried out was of some advantage, they could indicate it to reinforce the importance of the origin of the keratin.

• The authors indicate that they performed inhibition tests on the fungus Trichophytan rubran, however, I did not find this species, I found Trichophyton rubrum, they should review and correct if there are any errors.

• I suggest that within the introduction the authors indicate the importance of using the species they put in the antimycotic activity test since it is important to make it clear that Trichophyton rubrum (if it is the species they worked with) is a more anthropophilic dermatophyte. common throughout the world and its area of ​​infection include not only the feet but also the skin, hands, and nail plate.

• Doubt, why make a functional fabric? And not make some kind of film supplemented with its formulation to be used as dressings, or what is the advantage of the functional fabric?

Author Response

Reviewer 1: The article Therapeutic textiles functionalized with keratin-based particles encapsulating terbinafine for the treatment of Onychomycosis is a very interesting proposal that would help a very common problem. He only sent some comments and doubts.

1 - Review the writing because a text repeatedly appears indicating that there is an error.

Reply: The text was carefully reviewed and all the errors were resolved.                                   

2 - Because the authors carried out keratin extraction from hair and did they not use a reactive grade compound (to ensure composition)? It may be that the process that the authors carried out was of some advantage, they could indicate it to reinforce the importance of the origin of the keratin.

Reply: Changes were made according reviewers suggestion. Please see – Page 2 and 3, Lines 98-102.

3 - The authors indicate that they performed inhibition tests on the fungus Trichophytan rubran, however, I did not find this species, I found Trichophyton rubrum, they should review and correct if there are any errors.

Reply: Changes were made according to reviewer suggestion. Please see – Page 1 – Lines 21 and 27; Page 10 – Lines 291, 295, 303 and 310; Page 11 – Line 318; and Page 14 – Lines 434, 446 and 476.

4 - I suggest that within the introduction the authors indicate the importance of using the species they put in the antimycotic activity test since it is important to make it clear that Trichophyton rubrum (if it is the species they worked with) is a more anthropophilic dermatophyte. common throughout the world and its area of infection include not only the feet but also the skin, hands, and nail plate.

Reply: Information was included on Introduction section according to reviewer suggestion – Page 1 and 2 Lines 42 - 47.

5- Doubt, why make a functional fabric? And not make some kind of film supplemented with its formulation to be used as dressings, or what is the advantage of the functional fabric?

Reply: We thank the reviewer for the question. We plan to develop in  future work other strategies like films and nail lacquers, containing the keratin-based particle encapsulating terbinafine here developed. Information was added to the manuscript – Page 3 – Line 107 and Page 14, Lines 476-478.

Reviewer 2 Report

Dear Authors, 

The aim of the “Therapeutic textiles functionalized with keratin-based particles encapsulating terbinafine for the treatment of Onychomycosis” study is well explained, describing the design, physicochemical and biological properties of keratin-based carriers encapsulating terbinafine.

The paper is well organized and interesting to read, providing impressive information in each section, which is also sustained with explicit and well-illustrated graphics/ figures and tables. I recommend that the authors correct the names of the substances in the whole manuscript and improve the quality of the figures (for example, figure 4 has titles in different formats and the graphs are at low resolution). Also, I suggest interpreting the data results from a statistical point of view (like ANOVA) to apply all analyses performed in triplicate.

I highly recommend that the authors continue this study with the cytotoxicity evaluation of this carries on human cells and adjust the concentration of terbinafine in order not to develop drug resistance.

In my opinion, the manuscript could be published after minor revisions.

Author Response

Reviewer 2: Dear Authors, the aim of the “Therapeutic textiles functionalized with keratin-based particles encapsulating terbinafine for the treatment of Onychomycosis” study is well explained, describing the design, physicochemical and biological properties of keratin-based carriers encapsulating terbinafine. The paper is well organized and interesting to read, providing impressive information in each section, which is also sustained with explicit and well-illustrated graphics/ figures and tables.

1 - I recommend that the authors correct the names of the substances in the whole manuscript.

Reply: The name of the substances was reviewed and corrected along the manuscript.

2 - Improve the quality of the figures (for example, figure 4 has titles in different formats and the graphs are at low resolution)

Reply: Quality of the figures was improved. Titles of Figure 4 were uniformed and the quality of the figure was improved – Figure 4, Page 6

3 - Also, I suggest interpreting the data results from a statistical point of view (like ANOVA) to apply all analyses performed in triplicate.

Reply: The results were analyzed from a statistical point of view. Information was added to the manuscript – Pages 4, Lines 149-152; Page 4 – Figure 2 caption; Page 10 – Figure 8 caption; and Page 14 – Lines 458-461.

4 - I highly recommend that the authors continue this study with the cytotoxicity evaluation of this carries on human cells and adjust the concentration of terbinafine in order not to develop drug resistance.

Reply: We thank for the Reviewer for the excellent suggestion. We plan to follow up with these studies by testing the cytotoxic and genotoxic effect of the keratin-based particles encapsulating terbinafine on different human cells (keratinocytes and fibroblasts). Moreover, we also plan to evaluate the transungual drug delivery of encapsulated terbinafine on nails (healthy and infected).